# Social and Emotional Learning and Internalizing Problems among Adolescents: The Mediating Role of Resilience

**DOI:** 10.3390/children9091326

**Published:** 2022-08-31

**Authors:** Aurora Adina Colomeischi, Andreea Ursu, Ionela Bogdan, Alina Ionescu-Corbu, Romina Bondor, Elisabetta Conte

**Affiliations:** 1Faculty of Educational Sciences, Ștefan cel Mare University of Suceava, 720229 Suceava, Romania; 2“Riccardo Massa” Department of Human Sciences for Education, University of Milano-Bicocca, 20126 Milan, Italy

**Keywords:** social and emotional learning, internalizing problems, resilience, adolescents’ mental health

## Abstract

(1) Background: The high rates of mental disorders in adolescence presented in the literature often exclude internalizing problems. Although there is extensive data on the effectiveness of SEL skills in improving resilience, few studies included evidence in their reports on the relationship between SEL skills and internalizing problems. The present study aims to deepen the understanding of the relationship between SEL, resilience, and internalizing problems, by investigating the mediating effect of resilience between SEL components and internalizing problems. (2) Methods: Adolescents (N = 968 adolescents, aged between 11 and 18 years old; M = 13.30; SD = 1.92) from 30 schools from the NE region of Romania were invited to fill out questionnaires on social and emotional learning, internalizing problems, and resilience. (3) Results: The results show that resilience mediates the link between self-awareness and internalizing problems, between self-management and internalizing problems, between relationship skills and internalizing problems, and between responsible decision-making and internalizing problems. (4) Conclusions: These findings revealed the need for social and emotional learning interventions that include resilience-oriented approaches in order to decrease internalizing problems in adolescents. Moreover, we suggest that more culturally appropriate interventions are required to better investigate the interaction between SEL components, resilience, and internalizing problems.

## 1. Introduction

Adolescence is a vulnerable period in terms of development since, at this age, primary physical, psycho-social, and neurodevelopmental changes take place. Adolescents have the highest prevalence of mental disorders [1,2], which could lead to adverse outcomes in adulthood [3]. Extensive research uses the Achenbach’s classification of mental disorders, classifying them as internalizing and externalizing problems [4,5]. The internalizing problems include stress, anxiety, depression, social withdrawal, and psychosomatic symptoms, which can lead to eating disturbances, drug abuse or suicide attempts [6,7]. While externalizing problems have been intensively explored [8,9], less attention has been accorded to internalizing problems [10]. Internalizing problems can be more difficult to identify compared to the externalizing problems, which are more disruptive to the classroom and are the problems educators are more likely to address [11]. However, research consistently presents the magnitude of anxiety and depression among adolescents. For example, Barker and colleagues analyzed 37 studies of which 19 reported adolescents’ anxiety, and 36 reported adolescents’ depression [12]. Their results showed a prevalence of 19.1% for anxiety and 14.3% for depression, respectively. Moreover, the results of a recent study showed, in a large sample of 229,129 adolescents aged between 13 and 17 years, a prevalence of 17% for suicidal attempts, 48% being boys and 52% girls [13]. Past research also provided evidence for three categories of factors for internalizing problems: (a) individual factors such as poor self-esteem [14], low emotional intelligence [15], rumination [16], and low frustration tolerance [17]; (b) family factors such as the conflict between the parents, the absence of affection and support, as well as coercive authority [7]; and (c) contextual factors such as low income [18]. Nevertheless, the role of interpersonal and intrapersonal aspects, such as social and emotional learning (SEL), is often overlooked. Thus, we address this gap and explore the impact of SEL and its dimensions on internalizing problems.

### 1.1. SEL and Internalizing Problems

Social and emotional learning refers to the child’s ability to understand their emotions, to regulate them, and to engage in positive behaviors with their peers [19]. It is also a process through which children learn how to make good decisions and act responsibly [20]. These skills benefit young children even from the beginning of school as it helps them better accommodate school life and teachers. Such abilities are also linked to well-being in adult life [11]. School-life consists of social interactions with peers and teachers, emotional management, and developing a set of abilities that will help the children beyond the school ground [21]. Mental health is an important topic to be taken into consideration, especially because of the consequences of unaddressed mental health issues that can also lead to academic failure and difficulties in social and emotional interactions [11].

Social and emotional learning (SEL) comprises five core competencies as defined by The Collaborative for Academic, Social, and Emotional Learning (CASEL) as self-awareness, self-management, social awareness, relationship skills, and responsible decision-making. Self-awareness is assessing one’s strengths and limitations by recognizing the emotions and thoughts that lead to behaviors [22]. Self-management encompasses the ability to work toward goals by managing emotions and thoughts. Social awareness is the ability to be empathic, to be aware of different perspectives, to understand social and ethical regulations in a social environment, and to know where to seek support. Relationship skills comprise creating and maintaining healthy relationships through conflict resolution, communication, and help-seeking abilities. The last component, responsible decision-making, pertains to making healthy personal and social choices, and understanding the consequences of one’s actions in promoting well-being in the self and in others. The self-management component of SEL is related to managing emotions, as well as fostering more positive emotions and thus, encouraging better engagement in the classroom [11,23]. Social- and self-awareness is linked to the children’s emotional knowledge [19]. Responsible decision-making is associated with prosocial behavior and fewer self-reported symptoms of internalizing and externalizing problems [24]. Overall, all the components of SEL are linked to better mental health and outcomes in school and adulthood. Social and emotional learning is linked to successful outcomes in school in both the short and long-term academically, socially, and emotionally [25,26]. Having SEL abilities predicts lower internalizing problems in children, as they have better social interactions with peers, and better non-confrontational problem-solving abilities [19]. Similar results show that SEL skills help children have a higher quality in-school life, a better relationship with teachers, a better school attendance, and fewer internalizing problems [27,28].

### 1.2. SEL and Resilience

A recent model incorporating experiential learning, adversity, and inherent factors states that individuals must experience adversities to build resilience, which helps them develop new skills, or adapt previously developed skills to the new situation’s requirements [29]. This results in more resilience in various personal and social domains by the interaction of intrinsic factors and experiential learning [29,30]. According to Luthar and colleagues, resilience consists of an individual’s attributes that incorporate various internal and external vulnerabilities and protective factors in dealing with an adverse context; this leads to positive adaptation [30]. Whether regarded as a skill or a personality trait, it is probable that both intrinsic and extrinsic factors of resilience would work along with SEL skills in reducing internalizing problems [31,32]. Extensive research has focused on SEL programs’ effectiveness in fostering resilience, promoting positive mental health, and overall positive cognitive, emotional, and behavioral outcomes [33,34], as well as on executive functioning in high- and low-risk groups [35].

### 1.3. Resilience and Internalizing Problems

Resilience plays an essential role in alleviating the development of internalizing and externalizing problems in adolescence, and maintaining good overall mental health [36,37,38,39]. More precisely, higher levels of resilience are associated with lower levels of depression [40,41]. Previous studies reported that resilience had a moderating effect between stress and anxiety, and stress and depression in an adult population [42]. These findings are consistent with other reports on adolescents, where resilience mediated the effects between personality traits and depression [43]. Moreover, according to Konaszewski and colleagues, resilience is essential in helping adolescents achieve well-being by reducing the focus on negative emotions [37]. Fritz and colleagues supported this assertion by showing that lower levels of resilience at an earlier age predicted higher distress later in adolescence [44]. Additionally, Hildebrand and colleagues reported more internalizing and externalizing problems in children and adolescents with lower levels of resilience [45]. Similarly, Aksoy and colleagues pointed out that adolescents exposed to parental violence exhibited lower internalization symptoms through resilience [46], while other studies showed that family-related resilience had a significant impact on preventing the development of internalizing problems [47,48]. Previous researchers have examined the mediating effect of resilience on internalizing and externalizing problems, and various other factors. For example, previous research showed resilience had a mediating effect between stressful life events and anxiety [49], between sleep and behavioral and emotional problems [50], and between trauma, and depression [51]. Previous studies reported beneficial outcomes facilitated by resilience on enhancing positive behaviors, healthy relationships with peers, and school-related outcomes in school children [35]. To our knowledge, no study has examined the mediating role of resilience between the SEL dimensions and internalizing problems.

There is a high prevalence of mental health problems in children (20–30%) [52], and many of them have multiple problems [53], which are inadequately treated or undetected [54]. Grazzani and colleagues recently showed that lower levels of resilience predicted higher levels of internalizing problems [55]. More specifically, resilience mediated the relationship between SEL and internalizing problems. Students with higher scores in SEL obtained higher scores in resilience, leading to fewer internalizing problems. Gulseven and colleagues found that higher prosocial skills in preschoolers were linked to better emotional regulation and thus, fewer internalizing problems. Prosocial behaviors facilitate emotional regulation in young children. The authors also describe resilience as a protective factor in children that learn prosocial behaviors, as they have fewer internalizing problems in the future [56]. Thus, our objective is to investigate the relationship between SEL and internalizing problems by analyzing the direct effect of each component of SEL on internalizing problems, and then assessing the mediating role of resilience in the relationship between each dimension of SEL and internalizing problems.

### 1.4. The Present Study

The present study aimed to deepen the understanding of the relationship between SEL and internalizing problems by testing whether resilience mediated the relation between the SEL dimensions (self-awareness, self-management, social awareness, relationship skills and responsible decision-making) and internalizing problems. In order to test this mediation, we hypothesized that all SEL dimensions (self-awareness, self-management, social awareness, relationship skills and responsible decision-making) will be positively associated with resilience (H1.1), and negatively associated with internalizing problems (H1.2); resilience will be negatively associated with internalizing problems (H.2); and resilience will mediate the relations between SEL dimensions: self-awareness (H3.1), self-management (H3.2), social awareness (H3.3), relationship skills (H3.4) and responsible decision-making (H3.5), and internalizing problems.

## 2. Materials and Methods

### 2.1. Participants

The initial sample of the present study consisted of 1108 adolescents. One hundred forty-two persons were excluded from the study because they had not completed the entire questionnaire (demographic questions and scales). Thus, the final sample consists of 968 adolescents (63.8% females), aged between 11 and 18 years old (M = 13.30; SD = 1.92) living in the Nord-Eastern part of Romania. Sixty percent (*n* = 576; aged between 11 and 14 years old) were enrolled at secondary level, while forty percent (*n* = 392; aged between 15 and 18 years old) were enrolled at high school level. In Romania, secondary level consists of the fifth to the eighth grades, while high school level consists of the ninth to the twelfth grades. The primary level, the secondary levels, and the ninth and tenth grades are part of compulsory general education. The participants were recruited from 30 schools from the NE region of Romania.

### 2.2. Procedure

Secondary and high school students were invited to complete the questionnaires presented below. All legal representatives (school directors and parents) consented to the students participating in this study. All study questionnaires were filled out online. The students and the legal representatives were informed that participation is voluntary, and that the participants had the right to withdraw at any time without justifying the decision.

### 2.3. Materials

*Demographic questions.* This section includes socio-demographic questions such as gender, age, and level of education.

*Social and emotional learning.* A Romanian translation of Social Skills Improvement System, Social Emotional Learning Edition Brief Scales—Student Form (SSIS-SELb-S) was used to assess the social and emotional learning skills of secondary and high school students [57]. Participants were asked to rate each item by thinking of themselves. This self-report scale was developed on the theoretical model of CASEL and consisted of 20 items organized into five dimensions: *self-awareness* (e.g., “I ask for help when I need it”), *self-management* (e.g., “I stay calm when dealing with problems”), *social awareness* (e.g., “I help my friends when they are having a problem”), *relationship skills* (e.g., “I worked well with my classmates”), and *responsible decision-making* (e.g., “I do the right thing without being told”). All items were rated on a 4-point Likert scale from 1 (not true) to 4 (very true). In the present study, Cronbach’s alphas indices were 0.49 for self-awareness, 0.56 for self-management, 0.62 for responsible decision-making, 0.70 for social-awareness, and 0.57 for relationship skills. For the composite scale, Cronbach’s alpha was 0.85.

*Resilience.* A Romanian translation of the Connor Davidson Resilience Scale (CD-RISC 10) was used to measure adolescents’ resilience [58]. This self-report scale comprises ten items, rated on a 5-point Likert scale, from 1 (not true at all) to 5 (true nearly all the time). An example of one of the items is “I am able to adapt when changes occur”. The value of Cronbach’s alpha was 0.84.

*Internalizing problems.* A Romanian translation of the Strengths and Difficulties Questionnaire (SDQ) was used to assess the students’ perceived strengths and difficulties [59]. This self-report scale consists of 25 items, organized in 5 dimensions, with five items in each dimension. There is the conduct problems scale, hyperactivity scale, emotional problems scale, peer relationships problems scale, and prosocial behavior scale. These five dimensions can be organized into three supra-ordinated factors: *internalizing problems* (10 items, addressing emotional symptoms with items such as “I worry a lot”, and peer relationship problems with items such as “I am usually on my own. I generally play alone or keep to myself”); *externalizing problems* (10 items, addressing conduct problems with items such as “I get very angry and often lose my temper,” and hyperactivity with items such as “I am restless, I cannot stay still for long”); and *prosocial behavior scale* (5 items, with items such as “I try to be nice to other people. I care about their feelings”). For the present study, only the internalizing problems dimension was used. All items were rated on a 3-point Likert scale from 1 (not true) to 3 (certainly true). In the present study, Cronbach’s alpha indices for the internalizing problems factor was 0.62.

### 2.4. Data Analytic Approach

First, we conducted preliminary analyses consisting of descriptive statistics and correlations between the study’s main variables using IBM SPSS 26. Then, the mediational analyses were conducted using the macro PROCESS 3.5 in SPSS, Model 4 with one predictor, one criteria, and one mediator. PROCESS is a free tool for SPSS and SAS that integrates several statistical approaches from validated statistical tools (e.g., SOBEL, MODMED). A mediation analysis aims to specify the degree to which one predictor/independent/causal variable influences one criteria/dependent variable via one or more mediator variables [60]. We used 5000 bootstrapped samples, and biases were corrected at 95% confidence intervals (CI) to calculate the indirect effect of the mediating variable on the relationship between the predictor and the criteria. If the CI of the indirect effect does not include zero, it indicates that the indirect effect is significant at *p* < 0.05 [61]. The total effect © shows the total influence of the predictor on the criteria, and the direct effect (c′) illustrates the influence of the predictor on the criteria when the effect of the mediating variable is controlled [60]. In our proposed models, the five dimensions of SEL were each considered, in turn, the predictors, while the internalizing problems constituted the criteria, and resilience the mediator.

## 3. Results

### 3.1. Sample Characteristics

On average, the results depicted in Table 1 show that the participants obtained high scores on SEL dimensions and resilience, which suggests that the adolescents had high abilities to comprehend and manage their emotions, as well as to interact well with their peers. More precisely, they easily asked for and offered help, as well as communicated and collaborated with others. The high resilience score suggests their high adaptability when facing stressful events. However, the adolescents’ moderate scores on internalizing problems suggest they had difficulties in dealing with challenging emotions and situations.

### 3.2. Preliminary Analyses

Table 1 depicts the means, standard deviations, and correlations for the study’s variables. In general, the means were medium for SEL dimensions (self-awareness, self-management, social awareness, relationship skills, and responsible decision-making) and resilience, while for internalizing problems they were relatively low. All SEL dimensions (self-awareness, self-management, social awareness, relationship skills, and responsible decision-making) were positively associated with resilience. Four dimensions (self-awareness, self-management, relationship skills, and responsible decision-making) were negatively associated with internalizing problems. Resilience was negatively associated with internalizing problems.

### 3.3. Main Analyses

#### 3.3.1. Path Analysis. The Total Effect of SEL Dimensions on Internalizing Problems

SEL Self-awareness negatively correlated with internalizing problems (r = −0.21, *p* < 0.001), And the total effect of self-awareness on internalizing problems was also negative and statistically significant (c_1_ = −0.15, *p* < 0.001).

SEL Self-management was negatively associated with internalizing problems (r = −0.29, *p* < 0.001) and its total effect on internalizing problems was also statistically significant (c_2_ = −0.18, *p* < 0.001).

SEL Social awareness was statistically insignificant related to internalizing problems (r = 0.01, *p* = 0.77 > 0.05). The total effect of social awareness on internalizing problems was also statistically insignificant (c_3_ = 0.006, *p* = 0.77 > 0.05). We chose to exclude social awareness from mediation analysis.

SEL Relationship skills negatively correlated with internalizing problems (r = −0.24, *p* < 0.001); its total effect on internalizing problems was significant and negative (c_4_ = −0.15, *p* < 0.001).

Lastly, SEL Responsible decision-making was negatively related to internalizing problems (r = −0.20, *p* < 0.001) and its total effect on internalizing problems was negative and statistically significant (c_5_ = −0.19, *p* < 0.001).

#### 3.3.2. Path Analysis. The Direct Effect of SEL Dimensions on the Resilience and the Direct Effect of the Resilience on Internalizing Problems

The *a*_1_
*path* from self-awareness to resilience was positive and significant (a_1_ = 0.73, *p* < 0.001), and the b_1_ path from resilience to internalizing problems was negative and statistically significant (b_1_ = −0.15, *p* < 0.001).

The *a*_2_
*path* from self-management to resilience was significantly positive (a_2_ = 0.71, *p* < 0.001) and the *b*_2_
*path* from resilience to internalizing problems was negative and significant statistically (b_2_ = −0.13, *p* < 0.001).

The *a*_3_
*path* from relationship skills to resilience was positive and significant (a_3_ = 0.70, *p* < 0.01). The *b*_3_
*path* from resilience to internalizing problems was negative and statistically significant (b_3_ = −0.14, *p* < 0.001).

The *a*_4_
*path* from responsible decision-making to resilience was positive and significant (a_4_ = 0.61, *p* < 0.01). The *b*_4_
*path* from resilience to internalizing problems was negative and statistically significant (b_4_ = −0.15, *p* < 0.001).

#### 3.3.3. Path Analysis. The Indirect Effect of SEL Dimensions on Internalizing Problems through Resilience

The indirect effect of SEL Self-awareness on internalizing problems through resilience was statistically significant (a_1_ × b_1_ = −0.11, CI [−0.1419; −0.0862] (see Figure 1, subfigure a).

The indirect effect of SEL Self-management on internalizing problems through resilience was statistically significant (a_2_ × b_2_ = −0.09, CI [−0.1190; −0.0695] (see Figure 1, subfigure b).

The indirect effect of relationship skills on internalizing problems through resilience was statistically significant (a_3_ × b_3_ = −0.10, CI [−0.1320; −0.0714] (see Figure 1, subfigure c).

The indirect effect of SEL Responsible decision-making on internalizing problems through resilience was statistically significant (a_4_ × b_4_ = −0.09, CI [−0.1200; −0.0667] (see Figure 1, subfigure d).

Overall, the total indirect effect was significant statistically (a × b = −0.17, CI [−0.2140; −0.1246]. The direct effect was statistically insignificant (c′ = −0.06, *p* = 0.09 > 0.05), so there was a total mediation of resilience between SEL and internalizing problems (see Figure 2). Complete details about total, direct, and indirect effects are included in Table 2 and Table 3.

### 3.4. The Alternative Models

We tested the alternative models to exclude the other variants of mediation. The first alternative model comprised resilience as the predictor, SEL as the mediator, and internalizing problems as the criteria; it was statistically insignificant (a = 0.27, *p* < 0.001, b = −0.06, *p* = 0.08 > 0.05, a × b = −0.02, *p* = 0.09 > 0.05). The second alternative model used resilience as the predictor, internalizing problems as the mediator, and SEL as the criteria, and its indirect effect was also insignificant (a = −0.16, *p* < 0.001, b = −0.06, *p* = 0.08 > 0.05, a × b = 0.01, *p* = 0.09 > 0.05).

## 4. Discussion

Previous research on the effects of SEL interventions among adolescents showed a reduction in mental health problems such as depression, anxiety, and internalizing problems [62]. In addition, evidence from a large cross-sectional study showed a negative link between SEL skills and internalizing problems as well as resilience and internalizing problems, and a positive link between SEL skills and resilience [55]. Thus, the main aim of the present study was to deepen the understanding of the relationship between these variables by analyzing the links between all SEL dimensions, resilience, and internalizing problems. First, we hypothesized that all SEL dimensions (self-awareness, self-management, social awareness, relationship skills, and responsible decision-making) would be negatively associated with internalizing problems, and positively associated with resilience. The results indeed illustrated negative associations between each SEL dimension and internalizing problems (except for SEL Social awareness) and positive associations between all SEL dimensions and resilience, demonstrating that the first hypothesis was partially confirmed. Our findings on the associations between SEL skills and internalizing problems are consistent with previous research [19,55], though to our knowledge, few studies have explored this relationship. We found that self-management had a significant negative effect on internalizing problems. Previous research has shown that self-management reduces negative emotions, such as anxiety, as it is a protective factor from developing internalizing problems [23]. In addition, responsible decision making was negatively associated with symptoms of internalizing problems [24]. Self-management skills help reduce negative emotions, such as anxiety, as it is a protective factor from developing internalizing problems. The effect of SEL on internalizing problems is also evidenced in Durlak and colleagues’ meta-analysis [63]. Analyzing 213 studies involving 270,034 adolescents, they showed that higher levels of SEL determine lower levels of internalizing problems, such as social withdrawal, stress, anxiety, and depression. SEL skills determine personal growth through a gradual change in the source of thoughts and behaviors, from external and contextual factors to internalized thoughts and significance, assuming one’s own decisions and behavior [64].

Our results on social awareness and internalizing problems are intriguing, but somewhat consistent with previous literature. For example, Salavera and colleagues reported a negative association between internalizing problems and the emotional support facet, and no significant associations for the other components of social skills [65]. Nevertheless, these findings suggest that other characteristics which we did not consider in the present study might be underlying this relationship, thus further investigations are needed.

Concerning the positive association between SEL dimensions and resilience, Hromek & Roffey suggest that SEL dimensions are complementary, dynamic, and improve resilience [66]. Additionally, previous research on SEL-related programs showed a significant positive effect on resilience [67]. Furthermore, lifelong learning can have an essential effect on resilience, compelling people to cope with unfavorable circumstances [68].

According to the second hypothesis, our outcomes illustrate a negative association between resilience and internalizing problems. It is in line with the results of previous studies attesting that higher resilience predicts lower scores of depression, anxiety, and other internalizing problems [69,70]. Several investigations highlighted the importance of resilience in combating internalizing problems. Summarizing 49 studies, Dray and colleagues illustrated that resilience reduces internalizing problems, stress, and depressive symptomatology. and increases mental health [71]. The same authors suggest that an optimal level of resilience at the right age can improve interventions on internalizing problems.

However, mixed results are presented concerning the effect of SEL on internalizing problems over time. Cramer analyzed the impact of SEL on internalizing problems using a longitudinal study, and the results suggest that SEL increases resilience, but does not reduce the internalizing problems [72]. Instead, Siqueira and colleagues presented a negative effect of SEL on internalizing problems six months after an SEL intervention program ended [73].

The central hypothesis of the present study was that resilience mediates the relationship between SEL dimensions and internalizing problems. According to our findings, SEL dimensions were linked to resilience. Thus, self-awareness, self-management, relationship skills, and responsible decision-making were associated with the capacity to adequately adapt to negative contexts. In turn, resilience had an adverse association with adolescents’ internalizing problems such as stress, anxiety, depressive symptoms, somatic complaints, and social withdrawal. Similar reports were recently shown. Students with higher scores in SEL obtained higher scores in resilience, leading to fewer internalizing and externalizing problems [55]. The ability to respond adequately to adverse contexts explains the relationship between the capacities to recognize the emotions and thoughts that lead to behaviors, to work toward goals, to create and maintain healthy relationships, to make personal and social healthy choices, and to understand actions’ consequences, and internalizing problems such as stress, anxiety, depression, somatic disorders, and social withdrawal.

The findings of the present research have several practical implications. Previous research showed that the more frequently depression episodes emerged in adolescence, the more the chances of negative outcomes in adulthood increased [74]. Thus, the promising results of our study may prove that interventions aiming to improve SEL might prevent the onset of internalizing problems through resilience, which in turn could curb unfavorable developments later in adulthood. In addition, our results showed no link between the social awareness component of SEL and internalizing problems, which could suggest the need for more culturally sensitive approaches of SEL and resilience-oriented interventions. These data have potential implications for working towards more contextually-fit models of SEL and resilience in relation to internalizing problems. While other researchers found that SEL positively impacted resilience [75], but reported no significant results on internalizing problems [25,75], we found significant associations between SEL, resilience, and internalizing problems. Additionally, we found resilience totally mediated the relationship between SEL and internalizing problems, demonstrating the essential role resilience would play in future programs aimed at decreasing internalizing problems, hence advocating for considering resilience as a separate factor in future studies’ designs. Our findings also suggest it might be beneficial for school counselors and teachers to take into consideration the role resilience plays in the relationship between SEL components and internalizing problems, especially since the latter is generally less visible and reported.

These findings contribute to the adolescents’ mental health literature in several ways. First, the results of the current study brought additional knowledge by showing the associations between SEL dimensions, resilience, and internalizing problems. Second, we tested the mediating effect of resilience between four out of five SEL dimensions and internalizing adolescents’ problems. Third, we focused on one Eastern-European culture, Romania, for which there is no previous data. Fourth, we collected data from a large sample from the general population of adolescents.

In addition to its strengths, several limitations should be discussed. One of the most significant flaws is the cross-sectional design, which prevents us from knowing how these variables’ effects change over time, and from making causal assumptions [76]. Most research on SEL and internalizing problems is longitudinal and includes interventions to increase social and emotional skills. However, schools are a social and emotional environment, and learning is also a social process [20]. Despite having a cross-sectional design, we evidenced that the relationships between SEL dimensions and internalizing problems can be explained through resilience. Second, the instruments were self-administered, and desirability may have interfered. Future studies should collect data from various sources, like students, parents, and teachers.

## 5. Conclusions

The findings of the current study helped to establish resilience as a strong protective factor, especially in lowering levels of internalizing issues among adolescents. Although internalizing problems can be more challenging to identify compared to the externalizing problems that are more disruptive to the classroom, educators have the ability to address them.

Schools present essential social and emotional elements of learning [20]. Furthermore, Durlak and colleagues concluded in their meta-analysis that SEL programs could be efficiently integrated into regular school practices, as teachers are capable of implementing them, so that SEL can become an ingrained part of the school [63]. Thus, anxiety disorders, depressive disorders, social withdrawal, and somatic complaints among adolescents could be reduced in an inherent way [11]. SEL skills serve as a protective factor, making people more resilient and better equipped to deal with stressors and challenges throughout their lives [77]. Nevertheless, our mixed findings on the relationship between some SEL components and internalizing problems might signal the presence of other factors which we did not consider in the present paper. This may point to the need for more culturally-relevant evaluations of SEL and SEL interventions targeted to improve resilience and decrease internalizing problems.

## Figures and Tables

**Figure 1 children-09-01326-f001:**
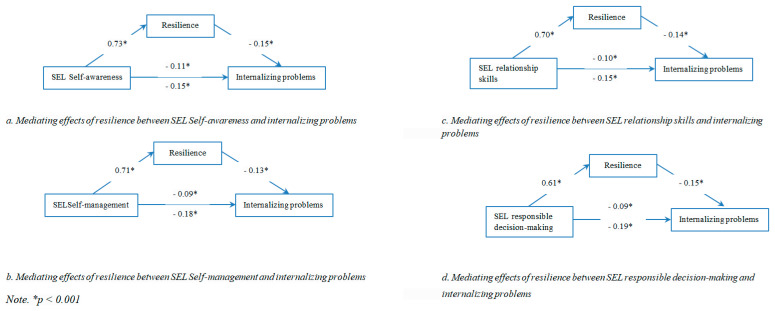
Mediating effects of resilience between SEL dimensions and internalizing problems.

**Figure 2 children-09-01326-f002:**
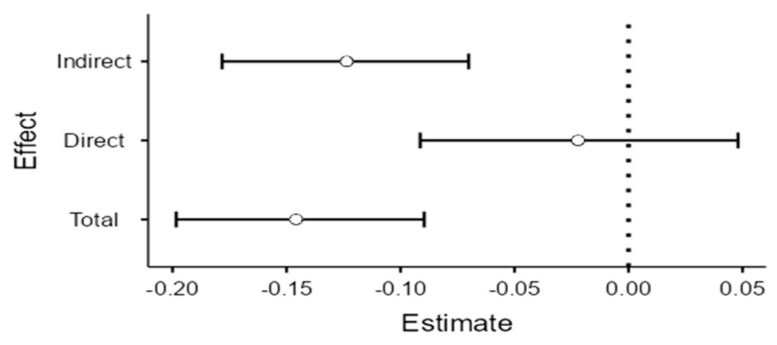
The mediation effect of resilience between SEL and internalizing problems: estimate plot.

**Table 1 children-09-01326-t001:** Means and correlations between study’s variables.

	M	SD	1	2	3	4	5	6	7	8
1. SEL self-awareness	3.30	0.45	-							
2. SEL self-management	3.06	0.49	0.55 *	-						
3. SEL social awareness	3.45	0.46	0.41 *	0.40 *	-					
4. SEL relationship skills	3.52	0.41	0.52 *	0.49 *	0.47 *	-				
5. SEL responsible decision-making	3.46	0.41	0.50 *	0.53 *	0.43 *	0.52 *	-			
6. SEL total	3.36	0.34	0.78 *	0.79 *	0.72 *	0.78 *	0.77 *	-		
7. Resilience	3.69	0.69	0.47 *	0.50 *	0.37 *	0.41 *	0.35 *	0.56 *	-	
8. SDQ internalizing problems	0.84	0.30	−0.21 *	−0.29 *	0.009	−0.24 *	−0.20 *	−0.37 *	−0.25 *	-

Note: * *p* < 0.001.

**Table 2 children-09-01326-t002:** Path estimates.

						95% Confidence Interval			
			Label	Estimate	SE	Lower	Upper	Z	*p*
SEL_1_	→	REZ	a_1_	0.7329	0.0460	0.6454	0.8256	15.92	<0.001
REZ	→	INT	b_1_	−0.1532	0.0167	−0.1867	−0.1198	−9.17	<0.001
SEL_2_	→	REZ	a_2_	0.7064	0.0404	0.623	0.7836	17.49	<0.001
REZ	→	INT	b_2_	−0.1323	0.0166	−0.165	−0.0997	−7.99	<0.001
SEL3	→	REZ	a_3_	0.6967	0.0518	0.594	0.7970	13.45	<0.001
REZ	→	INT	b_3_	−0.1434	0.0175	−0.178	−0.1095	−8.18	<0.001
SEL_4_	→	REZ	a_4_	0.6073	0.0551	0.498	0.7143	11.03	<0.001
REZ	→	INT	b_4_	−0.1510	0.0165	−0.184	−0.1186	−9.13	<0.001
SEL	→	REZ	a	1.1293	0.0576	1.016	1.24192	19.62	<0.001
REZ	→	INT	b	−0.1492	0.0189	−0.186	−0.11153	−7.90	<0.001

Note: SEL_1_ = SEL self-awareness, SEL_2_ = SEL self-management, SEL_3_ = SEL relationship skills, SEL_4_ = SEL responsible decision-making, SEL = total score of SEL dimensions, REZ = resilience, INT = internalizing problems.

**Table 3 children-09-01326-t003:** Direct, indirect, and total effects.

				95% Confidence Interval			
	Label	Estimate	SE	Lower	Upper	Z	*p*
Direct	c_1_	−0.0367	0.0242	−0.0828	0.0114	−1.51	0.130
Direct	c_2_	−0.0908	0.0209	−0.133	−0.0497	−4.34	<0.001
Direct	c_3_	−0.0869	0.0282	−0.142	−0.0311	−3.08	0.002
Direct	c_4_	−0.0631	0.0251	−0.112	−0.0147	−2.52	0.012
Direct	c	−0.0558	0.0326	−0.119	0.00799	−1.71	0.087
Indirect	a_1_ × b_1_	−0.1123	0.0136	−0.1404	−0.0859	−8.26	<0.001
Indirect	a_2_ × b_2_	−0.0934	0.0124	−0.119	−0.0700	−7.55	<0.001
Indirect	a_3_ × b_3_	−0.0999	0.0157	−0.133	−0.0711	−6.36	<0.001
Indirect	a_4_ × b_4_	−0.0917	0.0137	−0.120	−0.0659	−6.67	<0.001
Indirect	a × b	−0.1685	0.0230	−0.213	−0.12358	−7.33	<0.001
Total	c_1_ + a_1_ × b_1_	−0.1490	0.0221	−0.1924	−0.1047	−6.75	<0.001
Total	c_2_ + a_2_ × b_2_	−0.1842	0.0196	−0.223	−0.1452	−9.39	<0.001
Total	c_3_ + a_3_ × b_3_	−0.1868	0.0247	−0.236	−0.1380	−7.57	<0.001
Total	c_4_ + a_4_ × b_4_	−0.1548	0.0242	−0.202	−0.1069	−6.39	<0.001
Total	c + a × b	−0.2243	0.0280	−0.278	−0.16927	−8.01	<0.001

## Data Availability

Data supporting reported results can be requested from the corresponding author upon reasonable request.

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
