# Peer review of "Social and Emotional Learning and Internalizing Problems among Adolescents: The Mediating Role of Resilience"

_children, 2022, doi:10.3390/children9091326_

Round 1

Reviewer 1 Report

I would like to congratulate the authors for the study that they present.

1. Brief Summary:

The authors prudently conducted a research on an interesting subject. They examined the relation between SEL and the internalizing problems among adolescents, tested the mediation role of resilience of that relation. Their main findings showed that 4 of the 5 SEL dimensions (except social awareness) were negatively associated with internalizing problems and the resilience mediates the link between those 4 SEL dimensions and the internalizing problems.

2. Comments:

§ Introduction: the aim of the study is repeated in the final of the sub-heading 1.3 (lines 137-140) and in the beginning of the sub-heading 1.4 (lines 141-145). This must be revised to allow a logical and fluid reading, maybe changing a little the way of witting in the final of the sub-heading 1.3.

 §  Participants: the authors should clarify to the reader what means, in their country, the secondary level and the high school level, because there are some differences of the terminology worldwide.

 §  The sub-heading 3.1 “Data analytic approach” should be moved to the end of the chapter 2. “Materials and methods” and should be a little more detailed.

 §  The authors should start the chapter 3. Results with the sub-heading 3.1 Sample Characteristics, where they should describe the characteristics of the adolescents who participated in the study.

 §  In the line 229, the value of correlation is not correct: it is r = -.21 instead of r = -.22.

 §  In the line 242-243, the value of correlation is not correct: it is r = -.20 instead of r = -.29

 §  In the line 296 the authors make a reference to the Figure 2 but that Figure does not exist in the manuscript. This should be reviewed.

 §  The conclusions of the manuscript and the conclusions of the abstract should match in the main ideas, so the authors should improve the conclusions chapter.

 §  Most of the cited references have more than 5 years. I suggest the authors try to make some actualization of those.

 §  The STROBE Statement should be used by the authors for presentation of the results of the study.

Author Response

Dear Reviewer,

Thank you for your comments and suggestions.

Reviewer 2 Report

Dear authors, I congratulate the manuscript produced by you.

The subject addressed is quite interesting and necessary to interpret in view of the demands of the current knowledge society.

I really enjoyed reading your manuscript and I will make some comments that I hope will help to improve with a view to publication.

The introduction seems adequate and clear regarding the objective and hypotheses of the study.

The results are properly presented.

The discussion seems to be well-ordered, however I consider it to be quite summary. Authors can strengthen the discussion of hypotheses by contrasting with more recent studies that are available in the literature. Some points are only contrasted with a work that is not recent. An effort by the authors in this direction would be good to improve the quality of the manuscript.

The suggested limitations and research suggestions are appropriate.

The conclusions are obvious. However, given the value of the results obtained, what are their practical applications in the real context?

Best wishes for a good work.

Author Response

Dear Reviewer,

Thank you for your comments and suggestions.

Sincerely,

Adina Colomeischi
